# High Performance Carbon Material Prepared from Phalsa Using Mild Pyrolytic Process towards Photodegradation of Methylene Blue under the Irradiation of UV Light

**Sameerah I. Al-Saeedi** [1,*], **Muhammad Ali Bhatti** [2,*], **Aneela Tahira** [3], **Ghadah M. Al-Senani** [1], **Nada S. Al-Kadhi** [1], **Ayman Nafady** [4] and **Zafar Hussain Ibupoto** [5,*]

1   Department of Chemistry, College of Science, Princess Nourah bint Abdulrahman University, P.O. Box 84428, Riyadh 11671, Saudi Arabia
2   Institute of Environmental Sciences, University of Sindh, Jamshoro 76080, Pakistan
3   Institute of Chemistry, University of Sindh, Jamshoro 76080, Pakistan
4   Department of Chemistry, College of Science, King Saud University, Riyadh 11451, Saudi Arabia
5   Dr M.A Kazi Institute of Chemistry, University of Sindh, Jamshoro 76080, Pakistan
*   Correspondence: sialsaeedi@pnu.edu.sa (S.I.A.-S.); mali.bhatti@usindh.edu.pk (M.A.B.); zaffar.ibhupoto@usindh.edu.pk (Z.H.I.)

**Abstract:** In this study, we have used a mild pyrolytic process for the synthesis of luminescent carbon material from phalsa (Grewia asiatica Linn) and utilized it for the photodegradation of methylene blue (MB) in aqueous solution under the irradiation of ultraviolet (UV) light. The carbon material was found to be graphitic in nature and with carbon dot-like properties as demonstrated by powder X-ray diffraction (XRD), scanning electron microscopy (SEM), dynamic light scattering (DLS), and UV-visible techniques. The prepared carbon material was further studied for the elucidation of functional groups through Fourier transform infra-red (FTIR) spectroscopy. The carbon material exhibits the nanostructured phase which makes it a high surface area material for useful surface reactions. Different photodegradation aspects were investigated, such as initial dye concentration, catalyst dose, effect of pH of dye solution, reusability, electrochemical active surface area (ECSA), and charge transfer and scavenger. Optimum conditions of 15 mg carbon material, initial dye concentration of $2.3 \times 10^{-5}$ M solution, and pH 5 of dye solution gave the highest outperformance degradation efficiency. The degradation mechanism of MB in aqueous solution was dominated by the hydroxyl radicals as verified by the scavenger study. The reaction kinetics of MB degradation was followed by the pseudo first order kinetics and highest values of rate constants in the low initial dye concentration and the acidic pH of the MB solution. Significantly, the carbon material prepared from phalsa was found to be highly stable, as proven by the reusability experiments. Furthermore, the high ECSA and low charge transfer resistance of carbon material enabled it to have better performance. The use of mild pyrolytic process for the preparation of high performance luminescent carbon material from the biomass could be a great roadmap for the synthesis of a new generation of carbon materials for a wide range of applications including bio-imaging, catalysis, energy conversion and environmental applications.

**Keywords:** phalsa; carbon material; photocatalysis; methylene blue

## 1. Introduction

Today, the biomass-assisted preparation of nanoscale-based materials has been considered extensively. The unique aspect of using a biosynthesis process for the preparation of nanostructured materials is the feasibility of the chemical reaction using moderate reaction conditions and use of less toxic chemical compounds [1–3]. Different biomasses have been utilized, such as plant extracts, animals, microorganisms, viruses, DNA, and proteins for the synthesis of nanostructured materials [4–12]. The advantage of adapting

biosynthesis over physical and chemical methods is decided on the basis of an environment-friendly nature and negligible possibility of toxicity spillage during the synthesis. Furthermore, the inexpensive aspect from energy and pressure demand, high biocompatibility, stability and uniform dispersion in the polar solvents are the obvious features of biosynthesis [2,13–17]. Because of these advantageous features, biomass-assisted synthesis of nanomaterials has received potential use for the environment and energy generation processing [2,5,7]. The carbon dots at nanoscale were first reported in 2006 using carbon nanotubes through electrophoresis [18]. The carbon dots have two types of carbon atoms described by the sp2 and sp3 in the internal and external structure, respectively [19]. The surface of carbon dots is found to be highly hydrophilic, owing to the wide range of functional groups such as hydroxyl, carboxyl, epoxides, etc. which allowed them for being soluble in the polar solvents [20]. Recently, carbon dots are highly studied due to their attractive physical and chemical characteristics such as being chemically inert, having excellent luminescence, being very low toxicity, having high conductivity and being soluble in water [21]. These unique properties of carbon dots enable them to be utilized in diverse applications, including optical sensors, bio-imaging, electrochemiluminescence, solar cells, fuel generation [22], and photocatalysis [23]. The synthesis of carbon dots has taken place by two approaches: top-down and bottom-up. They are classified as pyrolytic or carbonization [24], chemical oxidizing agents-assisted [25], arch discharge [26], laser etching [27], microwave and solvothermal/hydrothermal process [28]. Recently, the biomass-based preparation of nanostructured carbon materials has received intensive attention; hence, a wide range of biomasses have been investigated for the synthesis of carbon nanostructures like chest nuts [29], coriander [30], lemon peel [31], cashew gum [32], honey [33], orange juice [34], orange peels [35], radish [36], egg [37], shrimp [38], pitahaya [39], milk [40], apple juice [41], stem of banana plant [42], pine apple peel [43], citric acid [44], corn and strawberry powder [45], rosemary leaves and biomass (glucose, chitin, and chitosan) [46]. They have been studied for different applications such as energy conversion, biological, and environmental applications [47]. The biomass-assisted synthesis of nanostructured carbon materials is low-cost, ecofriendly, has scale-up aspects, and is very facile; therefore, the new biomass-based synthesis of photocatalytic carbon is highly desirable in order to use them as alternative and efficient candidates for the wide range of applications including photocatalysis, energy conversion, bioimaging and biomedical [47]. The carbon dots have potential photoluminescence features through which they transform the visible light into low wavelengths; consequently, they have been utilized to enhance the optical band gap of semiconducting materials. Furthermore, the carbon-based photocatalytic materials have a high capability for the transfer and trapping of charge particles; hence, they are found to be excellent candidates for the photodegradation of synthetic dyes such as methylene blue (MB) [48]. The MB dye is very stable and very toxic for life and the environment, owing to its carcinogenic nature; hence, new and alternative technologies are required to mineralize MB into harmless products [48]. It has been highlighted that the efficient capability of photocatalysis process is shown in energy conversion, particularly using semiconducting compounds through the generation of electrons and holes during the irradiation of photons of light [49]. The carbon-based luminescent materials have a high ability to minimize the charge recombination of electrons and holes [50]. The degradation efficiency is related to reaction parameters including initial dye concentration and photocatalyst dose [51,52], pH of the dye solution [53], salt concentrations [54], and the location and nature of photons of light [55].

Keeping in view the importance of the biosynthesis process for the preparation of carbon-based photocatalytic material, and their potential applications towards the remedy of wastewater treatment and environment applications, we hereby report, for the first the time, the synthesis of carbon-based photocatalytic material from grewia, commonly known as phalsa. The phalsa is associated with the high density of polar groups, which are carried by a wide range of organic compounds including amino acids, carbohydrates and fatty acids; hence, it offers highly luminescent carbon material and easily dispersion in

polar solvents. Later, it can be used for a wide range of applications where we can have a challenge of homogenous dispersion of photocatalytic material. The phalsa fruit is not reported in the existing literature, owing to its occurrence throughout the world and is in high abundance and at a low-cost for the biosynthesis of versatile photocatalytic material.

In this study, we propose the simple, low-cost and large-scale production of photocatalytic carbon material from phalsa using a mild pyrolytic process on the collected juice of phalsa in limited air and at a slow rate of temperature. The photocatalytic properties of the newly prepared carbon materials were used for the photocatalytic degradation of methylene blue under the treatment of UV light photons. The structural studies have revealed the graphitic and carbon dot-like nature with enhanced photocatalytic properties for the degradation of MB with an efficiency of 99.32% in aqueous solution.

## 2. Results and Discussion

### 2.1. Morphology, Crystalline Structures, Optical Characterization and Functional Group Analysis of As-Prepared Carbon Material

The shape orientation of as-prepared carbon material from phalsa was studied by scanning electron microscopy (SEM) and the distinctive SEM images at different magnifications are shown in Figure 1. From the SEM results, it is obvious that the carbon material exhibits the typical sub-micron size and high degree of heterogeneity consisting rod, cube, and sheet-like morphology, which could be closely related to carbon dot features as show in Figure 1a,b. Functional groups were located on the surface of carbon material using FTIR analysis as shown in Figure 1c. Various vibrational bands were identified at $3439 \text{ cm}^{-1}$, $2926 \text{ cm}^{-1}$, $2850 \text{ cm}^{-1}$, $2520 \text{ cm}^{-1}$, $1730 \text{ cm}^{-1}$, $1631 \text{ cm}^{-1}$, $1431 \text{ cm}^{-1}$, $1018 \text{ cm}^{-1}$, $872 \text{ cm}^{-1}$, $788 \text{ cm}^{-1}$, $711 \text{ cm}^{-1}$, and $551 \text{ cm}^{-1}$. The vibration band of hydroxyl groups was located at $3439 \text{ cm}^{-1}$ on the surface of carbon material, whereas the stretching vibration of C-H bonds noticed at $2926 \text{ cm}^{-1}$ and SH vibration band observed at $2520 \text{ cm}^{-1}$. The banding vibrations of the C=O bond were identified at $1730 \text{ cm}^{-1}$, and the C-O and N-H bands were found at $1631 \text{ cm}^{-1}$ [56]. However, the stretching bands of C-N, N-N and –COO were indexed at $1431 \text{ cm}^{-1}$, and these bands were already found for the luminescent carbon dots related to their reducing properties [57–60]. The UV-visible absorbance spectrum of well-dispersed carbon in the water is shown in Figure 1d. A minor shoulder at 300 nm was observed for the n–π* transition of C=O and absorbance at 200 nm was also identified for the sp2 hybridized carbon from the aromatic ring connected to its π-π* [61]. A shoulder peak at 300 nm is identified by the unique C=O bonding sites where n-π* is the predominant transition as previously reported studies [62,63]. The obtained results from UV-visible analysis have verified the features of the as-prepared carbon material closely related to carbon dots [62,63]. The crystal quality of the as-prepared carbon material was illustrated by powder XRD, and measured reflections are shown in Figure 2a. There were two distinguished patterns at 19.58°, and 28.54° for the (001) and (002) crystal panes, respectively. These two patterns could be connected to the typical graphitic nature of carbon material [64], and the carbon material is weakly crystalline in nature, as described by XRD analysis. The aqueous solution of phalsa was placed in a quartz glass cuvette, illuminated with UV light for 10 min, and a mobile camera was used to take the camera image, as shown inset in Figure 2a. To understand the particle size distribution throughout the sample, we have performed a dynamic light scattering measurement, as shown in Figure 2b.The size of the distribution of carbon material was measured through Zetasizer Nano (ZS), with an average value of 515.7 nm. There were two distribution peaks, and the relative amount of particle size was quantified through the area of each distribution peak as shown in Figure 2b. The relative concentration of small particles of carbon was sufficient, suggesting the nanoscale size of as-prepared carbon material from phalsa, thereby exhibiting the large surface area which strengthened the enhanced photodegradation of MB in aqueous solution.

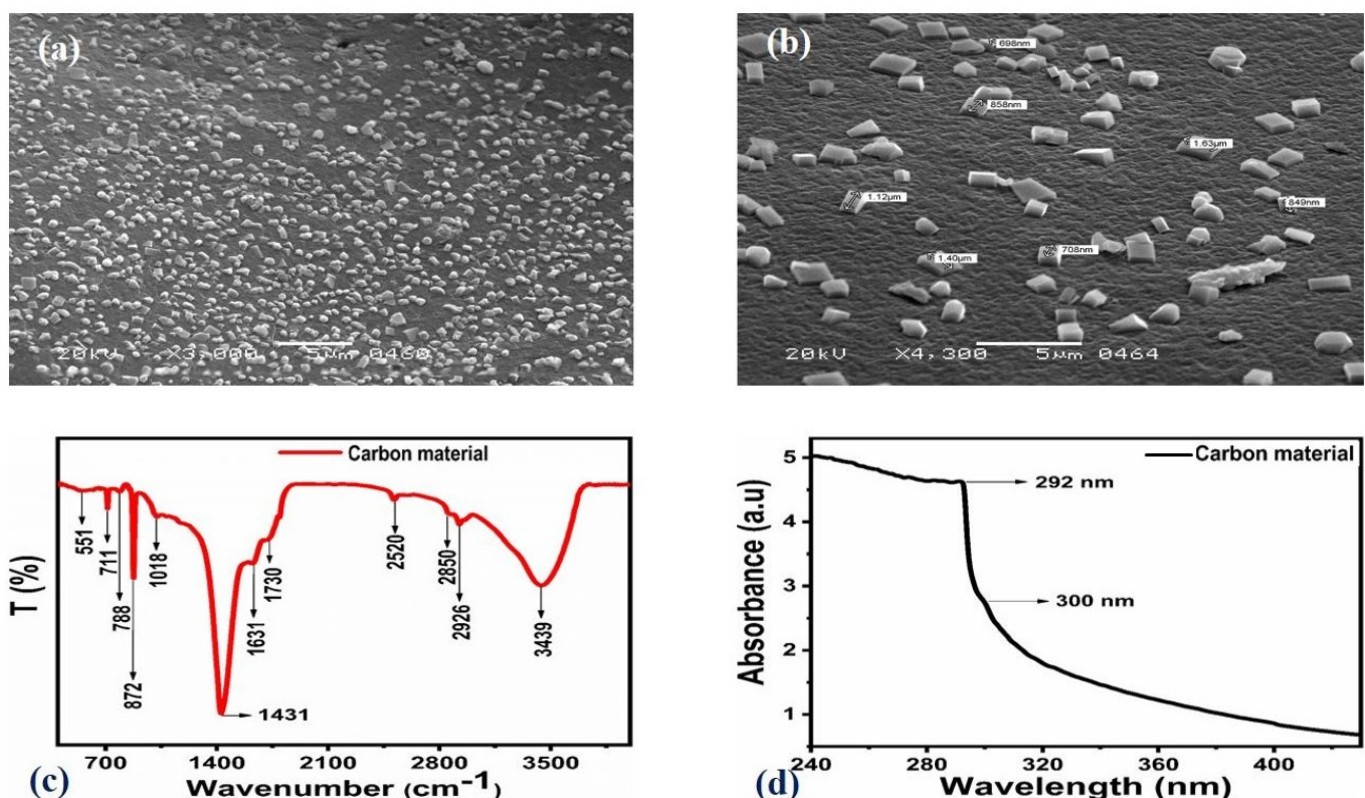

**Figure 1.** (**a**,**b**) Different magnified SEM images of carbon material from phalsa. (**c**) Functional group analysis of carbon material by FTIR. (**d**) Optical characterization of carbon using UV-visible spectroscopy.

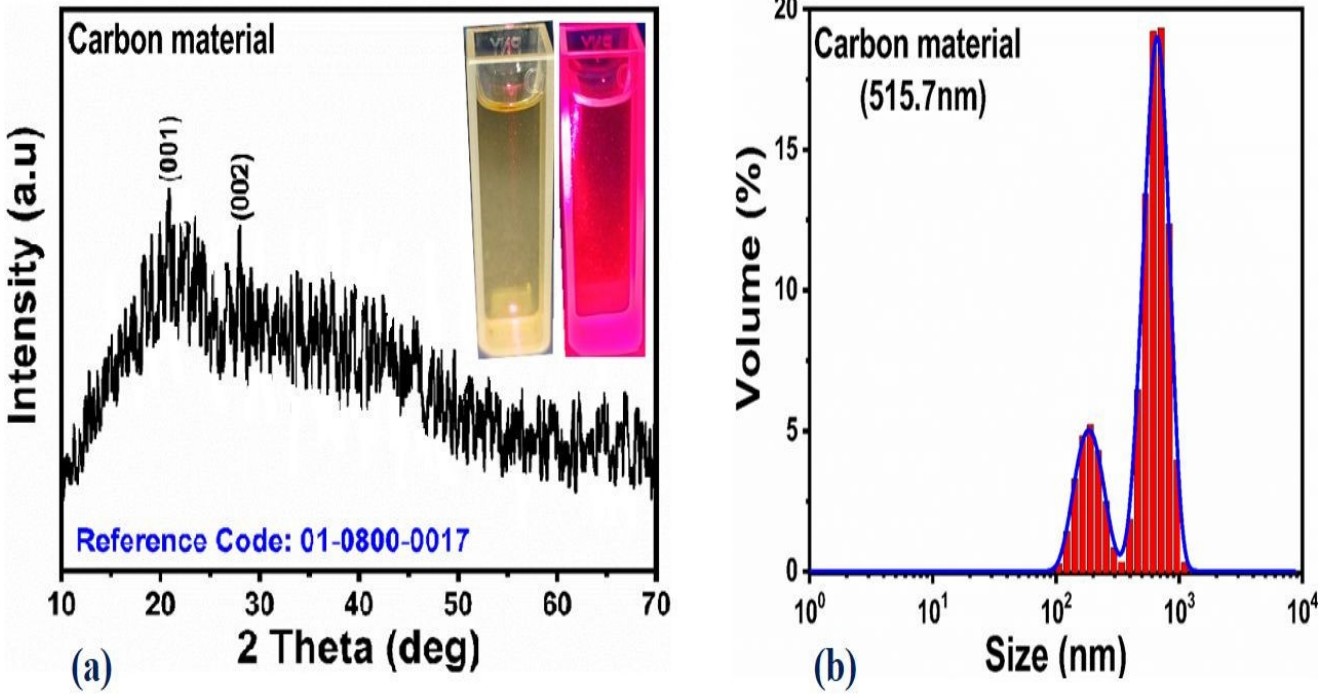

**Figure 2.** (**a**) Diffraction patterns of as-prepared carbon material from phalsa using powder; the inset represents a camera image of the luminescent properties of the as-prepared carbon material. (**b**) DLS analysis of as prepared carbon material using zeta nanosizer.

### 2.2. Photocatalytic Functionality of As-Prepared Photocatalytic Material

Before the evaluation of the photodegradation of MB in aqueous solution using newly-prepared photocatalytic material from phalsa, we have studied the degradation of $2.3 \times 10^{-5}$ M MB dye without carbon material and only using the exposure of UV light as shown in Figure 3. The absorbance changes reflect the negligible influence of UV light on the degradation of MB, indicating limited reaction kinetics as shown in Figure 3a. The kinetics parameters were also examined for the illustration of the reaction mechanism in terms of order of reaction as shown in Figure 3b,c. These analyses indicate that the rate constant value was found to be very low and UV light has only demonstrated a negligible role on the degradation process. The degradation efficiency of MB using only UV light exposure confirmed to be about 5–8%, defining the low performance of oxidation of MB as shown in Figure 3d. From the perspective of MB dye degradation using only UV light, it is very clear that we need the high performance of a new generation of photocatalysts which could be used immediately for the practical solutions of wastewater treatment challenges to avoid or minimize the polluting effects of dye exposure to the aquatic or our environment. For this purpose, we have used a low-cost, earth abundant, facile and scale-up methodology for the production of luminescent carbon material from phalsa using a mild pyrolytic process under limited air. We have investigated the photocatalytic performance of as-prepared carbon material from phalsa against the photodegradation of MB in aqueous solution under the irradiation of UV light. The photocatalytic performance of as-prepared carbon material was monitored in terms of the photocatalyst dose, initial dye concentration, reusability, effect of pH of dye solution, reusability and scavenger for the verification of the nature of radicals involved in the degradation process.

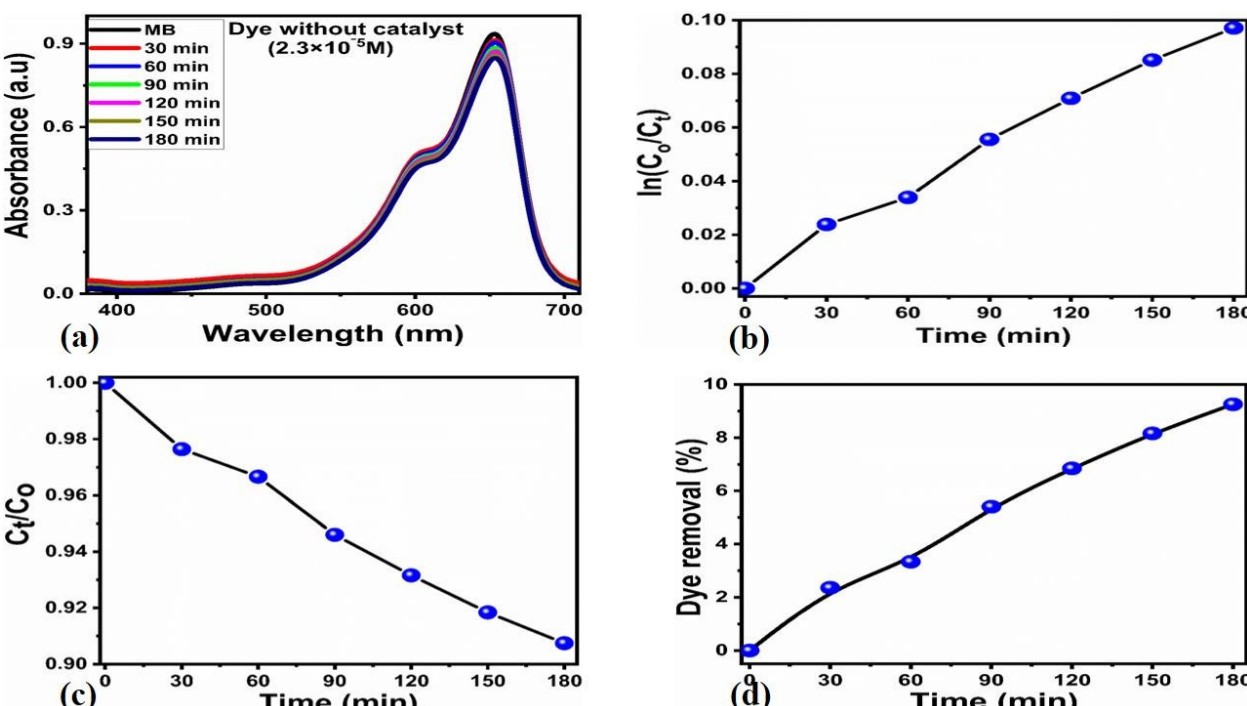

**Figure 3.** (**a**) UV-visible absorbance spectra measured in $2.3 \times 10^{-5}$ M MB solution in the absence of as-prepared carbon material and with the irradiation of UV for 180 min, (**b**,**c**) reaction kinetics of MB dye degradation without the photocatalyst, and (**d**) percent degradation efficiency with only the irradiation of UV light.

#### 2.2.1. Influence of Initial Dye Concentration on the Performance of As-Prepared Carbon Material from Phalsa

The initial dye concentration and catalyst role on the performance of carbon material were examined with two different concentrations of MB, including $2.3 \times 10^{-5}$ M and

$1.15 \times 10^{-5}$ M prepared in deionized water and irradiated by UV light. The obtained UV-visible absorbance spectra during photocatalysis were compared with the absorbance spectrum of bare MB solution before photocatalysis. Firstly, we have studied the photocatalytic activity of the as-prepared carbon material from phalsa in $2.3 \times 10^{-5}$ M under the influence of UV light, using a catalyst dose of 5, 10 and 15 mg for the time period of 180 min, as shown in Figure 4. Each absorbance change was examined with the time interval of 30 min, and it was found that the absorbance was successfully decreased with the time and increasing dose of catalyst as shown in Figure 4a–c. It was observed that the photocatalyst dose has verified the driving role during the photodegradation of MB, ad it has been previously studied by several reports [65]. This is because the high amount of photocatalyst dose offered more active site exposure to the frequent interaction with MB dye; hence, the effective degradation rate was observed with 15 mg of catalyst dose. Furthermore, the same dose of catalyst, including 5, 10 and 15 mg, was studied with the initial concentration of MB of $1.15 \times 10^{-5}$ M for the time period of 120 min and using UV light irradiation as shown in Figure 4d–f. Under the low concentration of MB, the rapid decrease in the absorbance is shown, suggesting the effectiveness of the carbon material prepared from phalsa and further revealing the initial dye concentration-dependent performance of carbon material as shown in Figure 4d–f. At the same time, it was also observed that the MB degradation is also highly dependent on the amount of catalyst dose. From the initial dye concentration and catalyst dose performance perspective, it is very clear that the MB degradation rate is governed by both initial dye concentration and catalyst dose. This analysis has verified that the optimum dye concentration and catalyst dose were $2.3 \times 10^{-5}$ M and 15 mg, respectively. In a high concentration of MB, the relative low performance could be connected to the thick layer of MB onto the surface of carbon material, and it exposed less surface for the interaction with the photons of UV light; hence, a smaller number of electrons and holes could be generated which further play a less effective role during MB dye degradation due to the limited number of produced oxidizing radicals causing the oxidation of dye.

### 2.2.2. Degradation of Kinetics of MB in Aqueous Solution Using Newly Prepared Carbon Material from Phalsa

The kinetics studies were performed for the photodegradation reaction of MB with the different concentrations, $2.5 \times 10^{-5}$ M and $1.15 \times 10^{-5}$ M, under the irradiation of UV light as described in Figure 5. The reaction kinetics of MB were studied with the mathematical equation $Ln(C_t/C_o) = kt$. In addition, the degradation rate was expressed by plotting $C_t/C_o$ against time. Both mathematical relations were used for 5, 10 and 15 mg of catalyst in $2.5 \times 10^{-5}$ M and $1.15 \times 10^{-5}$ M of MB concentrations, as shown in Figure 5a,b,d,e. In these relationships, $(C_t)$ and $(C_o)$ are the amount of the dye concentration left after certain time and the initial dye concentration (K) is the rate constant, and the (t) is the duration of photodegradation process. The calculated rate constant values in $2.5 \times 10^{-5}$ M and $1.15 \times 10^{-5}$ M for the 5, 10 and 15 mg of catalyst doses are shown in Table 1. It is clear that the rate constant values were strongly dependent on the initial MB dye concentration and catalyst dose. The low initial dye concentration with optimum catalyst dose of 15 mg have demonstrated the highest value of the rate constant as shown in Table 1. Moreover, the degradation efficiencies of 92.35%, 97.38% and 98.21% in $2.3 \times 10^{-5}$ M were found for 5, 10 and 15 mg catalyst doses, respectively, as shown in Figure 5c. The degradation efficiency of the 5, 10 and 15 mg catalyst dose against the $1.15 \times 10^{-5}$ M dye concentration were identified as 96.86%, 98.14% and 99.31% respectively as shown in Figure 5f. The rate constant value of the presented carbon material from phalsa with catalyst dose of 15 mg and high dye $2.5 \times 10^{-5}$ M shows the superior value, compared to previous reports on the same topic with an efficiency of 98.21% [66–69].

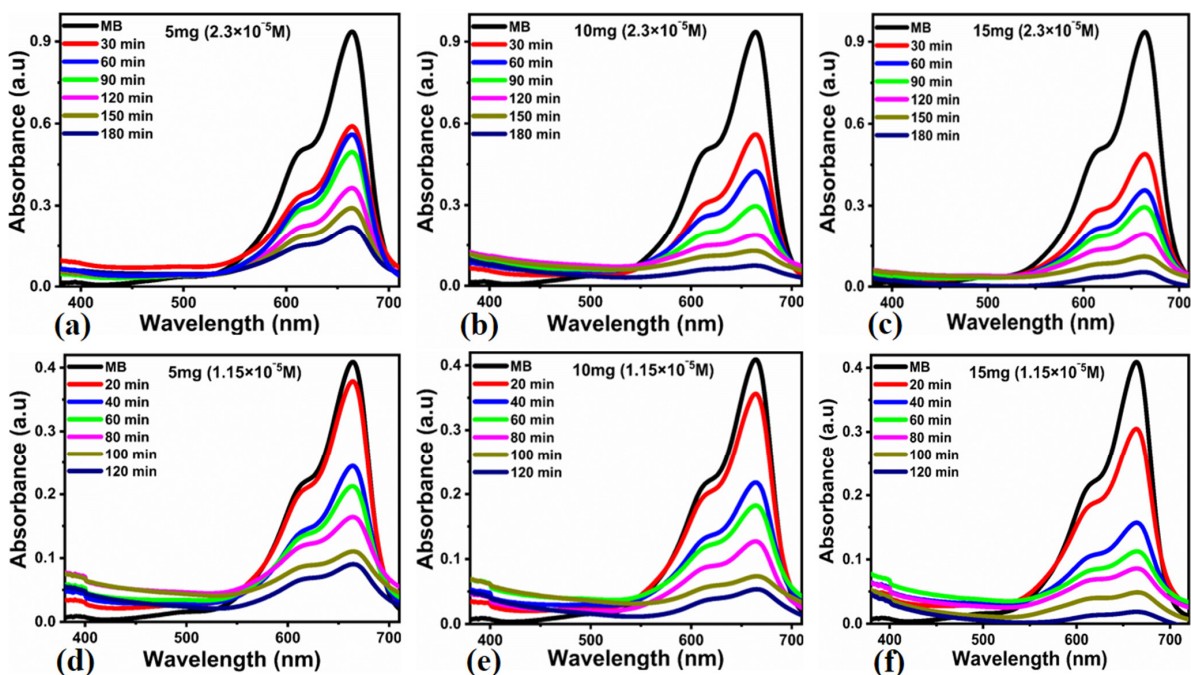

**Figure 4.** (**a**–**c**) UV-visible absorbance spectra measured in $2.3 \times 10^{-5}$ M MB solution using 5, 10 and 15 mg of as-prepared carbon material under the irradiation of UV for 180 min. (**d**–**f**) UV-visible absorbance spectra measured in $1.15 \times 10^{-5}$ M MB solution using 5, 10 and 15 mg of as-prepared carbon material under the irradiation of UV for 180 min.

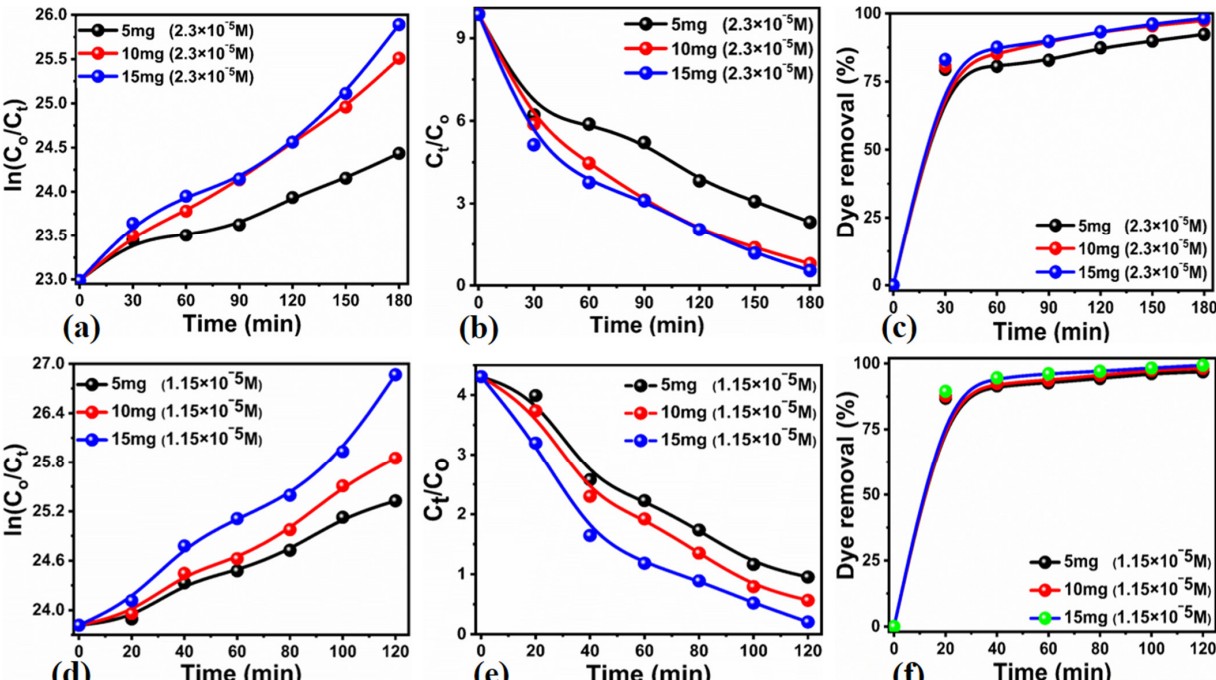

**Figure 5.** Degradation kinetics of MB using different catalyst doses of 5, 10 and 15 mg in $2.3 \times 10^{-5}$ M MB solution under the irradiation of UV light. (**a**) Linear plot of the natural logarithm of concentrations $\mathrm{Ln}(C_t/C_o)$ against time for 180 min. (**b**) Corresponding linear plot of $C_t/C_o$ (**c**) corresponding degradation efficiency MB solution. (**d**) Degradation kinetics of MB in $1.15 \times 10^{-5}$ M MB with a corresponding linear plot of the natural logarithm of concentrations $\mathrm{Ln}(C_t/C_o)$ against time for 120 min. (**e**) Corresponding linear plot of $C_t/C_o$ in $1.15 \times 10^{-5}$ M MB (**f**) corresponding to the degradation efficiency of $1.15 \times 10^{-5}$ M MB solution.

**Table 1.** Summarized performance evaluation of as-prepared carbon material from phalsa against the photodegradation of MB under the irradiation of UV light.

| Sample Dose | Dye Con: | Constant (K) | Dye Con: | Constant (K) | Scavengers | Constant (K) |
|---|---|---|---|---|---|---|
| 5 mg | $2.3 \times 10^{-5}$ M | $7.36 \times 10^{-3}$ min$^{-1}$ | $1.15 \times 10^{-5}$ M | $1.32 \times 10^{-2}$ min$^{-1}$ | $C_6H_8O_6$ | $1.33 \times 10^{-3}$ min$^{-1}$ |
| 10 mg | | $1.34 \times 10^{-2}$ min$^{-1}$ | | $1.73 \times 10^{-2}$ min$^{-1}$ | $NaBH_4$ | $1.17 \times 10^{-3}$ min$^{-1}$ |
| 15 mg | | $1.45 \times 10^{-2}$ min$^{-1}$ | | $2.39 \times 10^{-2}$ min$^{-1}$ | EDTA | $1.21 \times 10^{-3}$ min$^{-1}$ |
| pH study | | | | | | |
| pH-5 | $2.3 \times 10^{-5}$ M | | | $3.92 \times 10^{-2}$ min$^{-1}$ | | |
| pH-7 | | | | $2.79 \times 10^{-2}$ min$^{-1}$ | | |
| pH-9 | | | | $2.53 \times 10^{-2}$ min$^{-1}$ | | |
| pH-11 | | | | $2.16 \times 10^{-2}$ min$^{-1}$ | | |

### 2.2.3. Influence of pH on MB Dye onto the Photocatalytic Activity of As-Prepared Carbon Material

The role of pH on photocatalytic performance is highly important during the degradation as previously reported [65], due to the change in the nature of charge on the surface of photocatalytic material when it is immersed in a specific pH of dye solution. For this reason, we have also studied effect of pH in $2.5 \times 10^{-5}$ M MB dye solution using 15 mg of newly-prepared carbon material from phalsa under the irradiation of UV light as shown in Figure 6. We have adjusted the pH of 5, 7, 9 and 11 for the $2.5 \times 10^{-5}$ M MB solutions and illuminated each solution with UV light for the time period of 90 min. After the degradation of MB process, the initial pH of dye solution was changed slightly from 5, 6, 7 and 11 to 6.4, 8.9, 10.7, and 12.6, respectively. These changes in the pH after the dye degradation suggest that the different environments of different species produced results which were differently influenced on the changes in the pH of dye solution. All pH-dependent studies were performed at the room temperature and pressure as shown in Figure 6. Keeping the constant value of catalyst at 15 mg, and only varying the pH of MB dye solution has drastically affected on the degradation kinetics, suggesting that the new type of radicals generated play a leading role in the dye degradation process. It was found that the dye solution maintained at pH of 5 has the highest dye removal performance of 99.25%, whereas the degradation efficiency of 97.51%, 96.81% and 95.68% was observed for the pH 7, 9 and 11, respectively 7c. Acidic pH-dependent performance could be indexed to the higher concentration of hydroxyl radicals, thereby acting as a potential oxidizing agent towards the degradation of MB as previously reported [70]. It is previously reported that the MB degradation rate, with a superior performance in acidic condition, was linked to the high capability of dissolution of unprotonated MB molecules in the aqueous solution [71]. As a result of the interaction with holes generated during the irradiation of catalyst surface with UV light, hydroxyl radicals form, which later play an important role during the degradation of MB [71]. Additionally, it was found that the oxidation of hydroxide ions localized on the surface of carbon material have a leading contribution towards the degradation of MB via hydroxyl radicals [72]. Moreover, it has been known that the efficiency of photocatalysis at various pH values is mainly governed by the intrinsic nature of photocatalyst [73]. The effect of different pH values of $2.5 \times 10^{-5}$ M on the degradation of MB solution on the degradation kinetics was also studied. It was also observed that the pH of MB dye solution slightly varied for the pH values 5, 7, 9 and 11 to 4.80, 7.3, 9.6 and 11.8, respectively. In addition, the reaction kinetics of MB in $2.5 \times 10^{-5}$ M using 15 mg catalyst under the pH values of 5, 7, 9 and 11 were studied as shown in Figure 7a,b. Figure 7a shows the kinetics study of the MB degradation using different pH values of MB dye solutions. The reaction kinetics in the acidic conditions were fully following the pseudo order kinetics. The degradation efficiency was also shown

by the various pH values of MB dye solutions, suggesting that the dye degradation is highly favorable under the acidic pH as shown in Figure 7c. The corresponding rate constant values are shown in the Table 1, with highest rate constant value for pH 5 and such high values of rate constant are fully supported by published work [70].

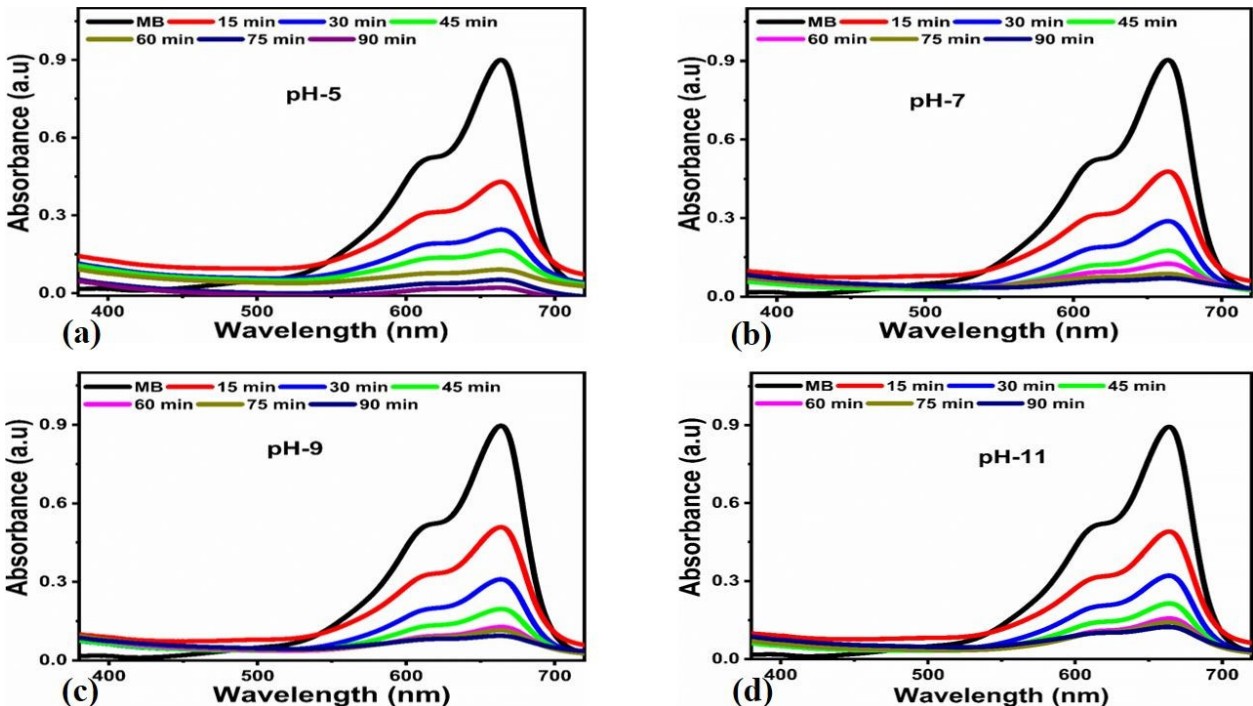

**Figure 6.** (**a–d**) UV-visible absorbance spectra measured in $2.3 \times 10^{-5}$ M MB solution with an adjusted pH of 5, 7, 9 and 11, using 15 mg of as-prepared carbon material under the irradiation of UV for 90 min.

### 2.2.4. Scavenger Study about the Verification Nature of Radicals Involved in the Degradation of MB

For an understanding of the degradation process of MB in aqueous solution, we have studied the degradation of MB under the influence of wide range of scavengers for the verification of the nature of radicals that predominate the efficient degradation of MB as shown in Figure 7d–f. Figure 7d shows the kinetics of the MB degradation under the influence of various scavengers. The selected scavengers include ascorbic acid, sodium borohydride, and ethylenediamine tetracetate (EDTA). Previously, it was shown that the radicals such as hydroxyl ($\cdot$OH), superoxide ($\cdot O_2$) and photon-assisted generated electron-hole pairs have been responsible for the degradation of MB. It is also shown that the mineralization of MB is followed by the involvement of oxidizing radicals such as ($\cdot O_2$) and hydroxyl ($\cdot$OH); hence, the selection of scavengers in the present study is based on them [74]. The efficiency of degradation is largely connected to the density of these oxidizing radicals, as shown in earlier studies [75,76]. Sodium borohydride has been found responsible for the suppression of degradation of MB owing to its strong influence on the generation of hydroxyl radicals; therefore, the present work has confirmed that the degradation of MB was mainly controlled by the participation of hydroxyl radicals, which is also confirmed by the already reported works [75]. The degradation kinetics, under the influence of scavengers, was followed by the pseudo first order kinetics as shown in Figure 7d–f and rate constant values were found to be the lowest for the degradation of MB under the environment of scavenger as shown in Figure 7. Rate constant values were found to be the lowest for the degradation of MB under the environment of scavengers as given in Table 2. Furthermore, the degradation efficiency was found to be lower compared to the

degradation efficiency without the use of scavengers, suggesting that hydroxyl radicals were largely passivized as shown in Figure 7f.

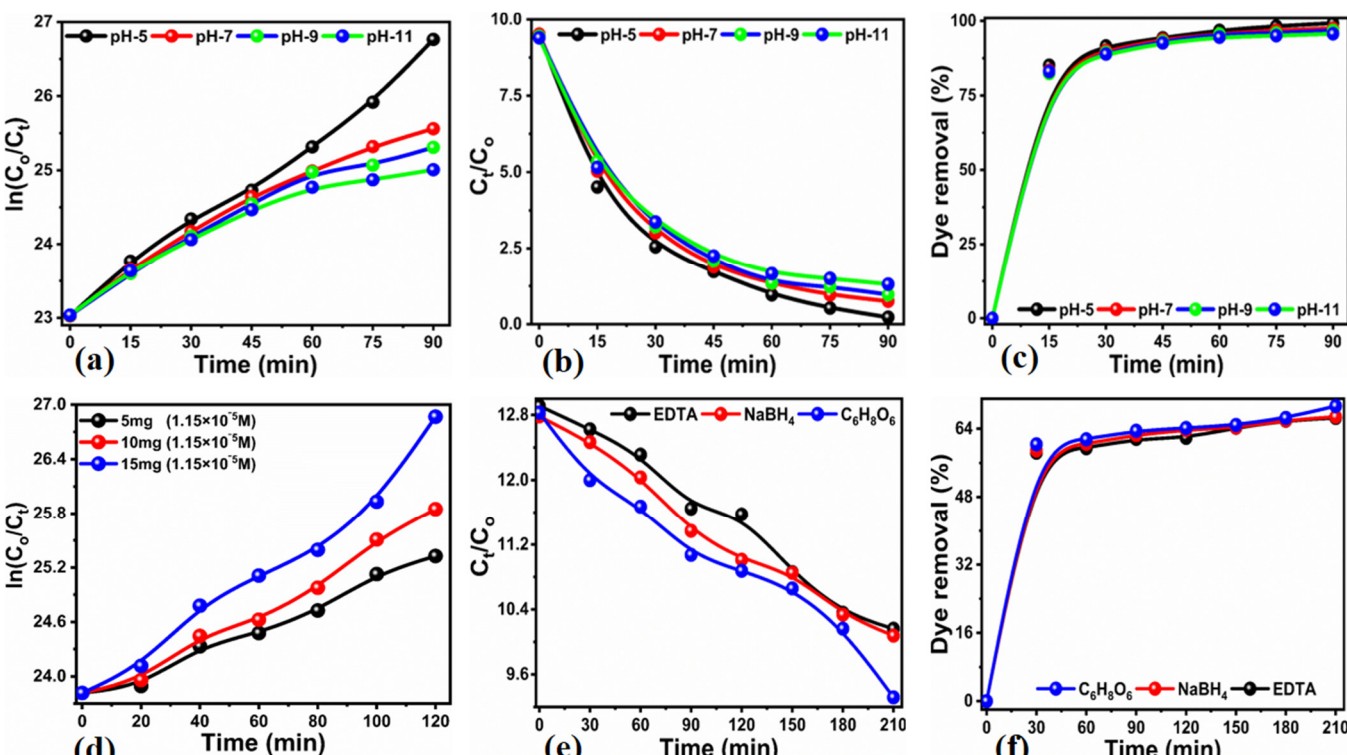

**Figure 7.** Degradation kinetics of MB using different catalyst doses of 15 mg in $2.3 \times 10^{-5}$ M MB solution under the irradiation of UV light. (**a**) Linear plot of the natural logarithm of concentrations $Ln(C_t/C_o)$ against time for 90 min. (**b**) Corresponding linear plot of $C_t/C_o$. (**c**) Corresponding degradation efficiency MB solution. (**d**) Degradation kinetics of MB in $2.3 \times 10^{-5}$ M MB with a corresponding linear plot of the natural logarithm of concentrations $Ln(C_t/C_o)$ against time for 120 min, in the presence of various scavengers. (**e**) Corresponding linear plot of $C_t/C_o$ in $2.3 \times 10^{-5}$ M MB in the presence of various scavengers. (**f**) Corresponding degradation efficiency of $2.3 \times 10^{-5}$ M MB solution in the presence of various scavengers.

## 2.2.5. Generalized Degradation Mechanism of Aqueous MB Solution Using Carbon Material from Phalsa

The irradiation of UV light onto as-prepared carbon material from phalsa with the MB dye solution and the degradation mechanism can be visualized from Scheme 1. In the generalized mechanism, the irradiation of UV light can generate a variety of oxidizing radicals like ·OH and ·$O_2$, which have been used for the degradation of MB in aqueous solution. There are several short-lived species have been produced which are spontaneously transformed into a wide range of harmless products like $CH_4$, $H_2O$, $SO_3$, $NO_2$, $CO_2$ and $NH_2$. The published work on the degradation of MB in aqueous solution is described by the dissociation of a homo and heterocyclic aromatic ring [70]. In the presented study, UV light irradiated newly-produced carbon material and possibly resulted in the generation of various radicals which take place in the photodegradation process of MB in aqueous solution. The irradiation of UV light photons with the surface of carbon material enables the provision of many electron-hole pairs with enough energy within the conduction or valence band structure. This leads to the efficient degradation of MB at the end. We tried to explore the photocatalytic properties of the as-prepared carbon material in the natural sunlight, but we found very little effectiveness compared to the UV light.

**Table 2.** Comparative results of the proposed study with the published analyzed data on pollutants.

| Catalyst | Dye Concentration | Light Source | Time (min) | Removal (%) | Synthesis Method | Ref. |
|---|---|---|---|---|---|---|
| $TiO_2$-MCDs | MB; 10 ppm | Visible light | 120 | 83% | Two-step microwave treatment method | [77] |
| $TiO_2$-CDs | MB; 10 ppm | Visible light | 60 | 40.9% | Hydrothermal | [78] |
| CDs/$TiO_2$ | MB; (1 mg/mL) | Visible light | 120 | 90% | One-pot Electrochemical method | [79] |
| CQD/$TiO_2$/$Fe_2O_3$ | MB; (20 mg/L) | Visible light | 180 | 86% | Multi-step hydrothermal | [80] |
| N-CQDs | MB; (20 mg $L^{-1}$) | Solar light | 260 | 97% | Hydrothermal carbonization | [81] |
| CQDs/$Cu_2O$ | MB; (50 mg $L^{-1}$) | UV light | 240 | 90% | One-step ultrasonic | [82] |
| M-CDs/$TiO_2$ | MB; (10 ppm) | Visible light | 240 | 83.0% | Two-step microwave treatment method | [83] |
| $TiO_2$-NCQD | MB; (10 mg $L^{-1}$) | Visible light | 420 | 86.9% | Hydrothermal Method | [84] |
| GQDs | MB; ($1.954 \times 10^{-6}$ mol/L) | Visible light | 120 | 79.4% | Heat treatment | [85] |
| photocatalytic carbon material | MB; ($1.15 \times 10^{-5}$ M) | UV light | 120 | 99.3 | Carbonization | Our work |

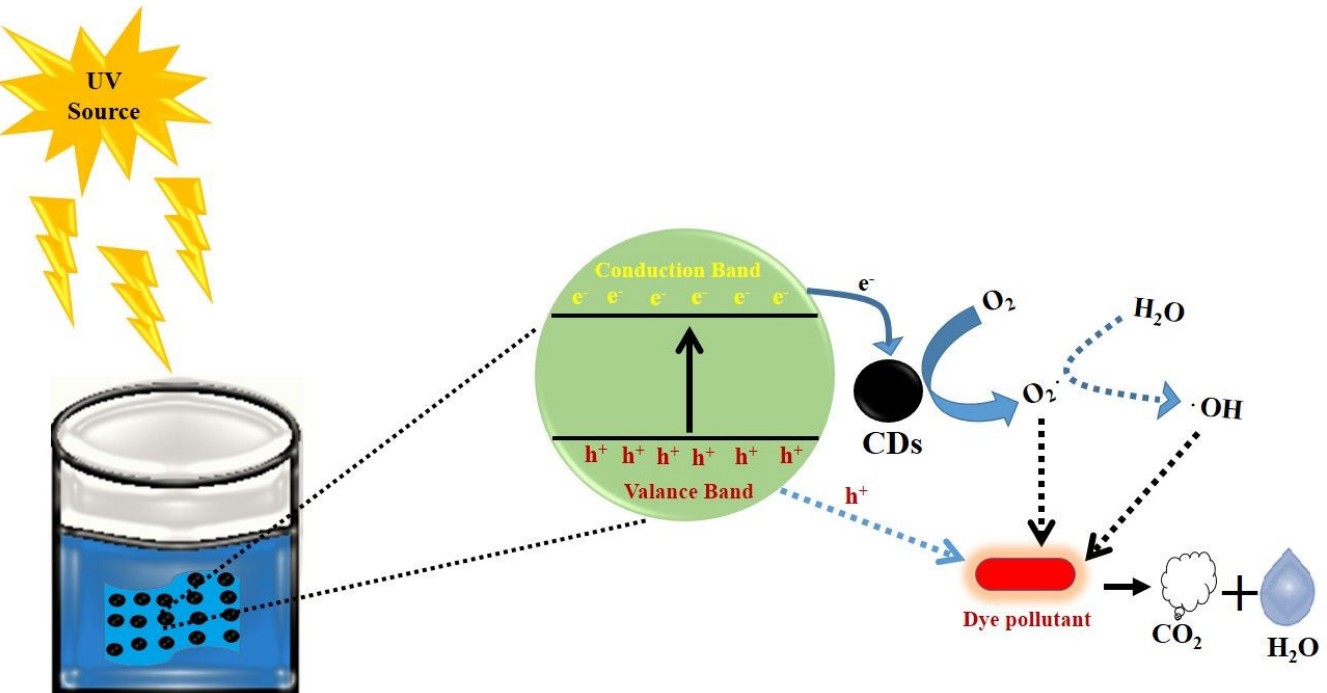

**Scheme 1.** Generalized reaction mechanism of MB on the surface of as-prepared carbon material from phalsa.

### 2.2.6. Reusability and Charge Transfer of As-Prepared Carbon Material during the Photodegradation of MB in Aqueous Solution

Three repeated cycles of as-prepared carbon material were used to examine the reusability aspects against the photodegradation of MB dye in $2.5 \times 10^{-5}$ M and using 15 mg catalyst as shown in Figure 8. Figure 8a describes the 1st, 2nd and 3rd cycle of degradation of MB using as-prepared carbon material under the irradiation of UV light. The carbon material before the recycling experiment was considered as a control measurement and was named a 0 cycle, as shown in Figure 8a. From this experiment, it was verified that the material performance remains very good, even after the use of three cycles, which could indicate an excellent stability of the material. The degradation efficiencies of the three cycles are presented as bar graphs as shown in Figure 8a. The degradation effectiveness was estimated to be 99.32%, 98.08%, 97.07% and 96.71% for the 0, 1st, 2nd and 3rd cycles of degradation of MB in aqueous solution. These findings suggest that such a high-performance carbon material for the photodegradation of MB under the irradiation of UV light can be used as an alternative material, owing to its green nature and being earth abundant, inexpensive and ecofriendly. Moreover, the reason for the outperform functionality of as-prepared carbon material from phalsa against the degradation of MB in aqueous solution was further understood by the charge transfer and electrochemical active surface area (ECSA) as shown in Figure 8b–d. The measurements of ECSA and EIS were performed as per our precious studies [76]. First, we tried to quantify the ECSA value from the non-Faradic region of the cyclic voltammetry curves measured at different scan rates as shown in Figure 8b. The linear fitting of the difference of current densities of anode and cathode sides of the CV curves and fitted data has given a slope, which corresponds to the value of ECSA as shown in Figure 8c. The estimated ECSA value was as found to be 4.32 mFcm$^{-2}$. We have also studied the EIS for the information of charge transfer to support the excellent degradation performance of as-prepared carbon material through the quantified value of charge transfer resistance as shown in Figure 8d. The experimental EIS data was simulated and well-fitted with an equivalent circuit as shown as inset in Figure 8d. The equivalent circuit was based on the elements such as solution resistance (Rs), carbon material resistance (Rct), and constant phase elements (CPE). The calculated charge transfer resistance of carbon material in MB solution was found to be 955.25 and 771.67 ohms for the carbon material in the deionized water and $2.5 \times 10^{-5}$ M dye solution, respectively. Interestingly, the solution resistance (Rs) was also measured for the deionized water and dye solution as 16.85 and 7.53 ohms respectively, confirming the favorable charge transfer during the degradation of dye under the irradiation of UV light [86,87]. Both the significant value of ECSA and low charge transfer resistance of as-prepared carbon material from phalsa strongly support the high-performance functionality against the degradation of MB under the irradiation of UV light. It is obvious that the proposed synthesis is low-cost, simple, scaled-up and ecofriendly towards efficient degradation of MB in a high concentration and low catalyst dose under the irradiation of UV light.

Moreover, it has been previously shown that the inorganic anions such as Cl$^-$, HCO$_3{}^-$, PO$_4{}^3$, SO$_4{}^{2-}$, etc., have been accompanied with the wastewater from the textile industry; hence, they have a significant effect on the retardation of MB dye oxidation. These kinds of studies have to be considered during the oxidation of organic dyes and the role of such inorganic anions should be explored in order to realize the new generation of photocatalysts for practical applications.

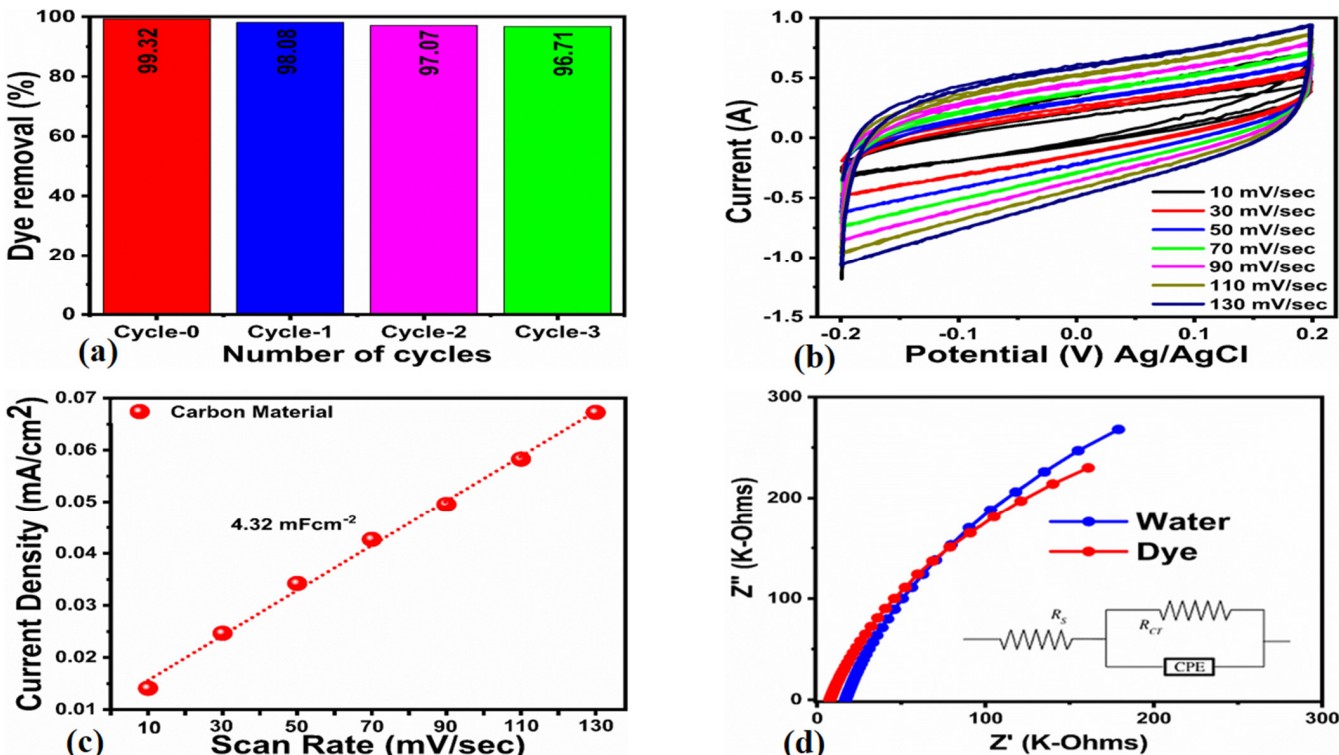

**Figure 8.** (**a**) Reusability of three cycles measured in $2.3 \times 10^{-5}$ M MB solution. (**b**) Cyclic voltammetry measured in $2.3 \times 10^{-5}$ M MB solution at various scan rates. (**c**) Linear fitting of difference anodic and cathodic current density for measuring (ECSA). (**d**) EIS measurement in $2.3 \times 10^{-5}$ M MB solution and bare water under the irradiation of UV light at a sweeping frequency rate of 100 kHz to 0.1 Hz, an amplitude of 10 mV and zero bias potential in open circuit.

## 3. Experimental Section

### 3.1. Chemical Reagents

Methylene blue, sodium hydroxide, hydrochloric acid (37%), ascorbic acid, sodium borohydride, ethanol, and disodium ethylenediamine tetracetate were of high analytical grade and purchased from Sigma Aldrich, Karachi Sindh, Pakistan. The deionized was used to prepare the desired solutions.

### 3.2. Preparation of Carbon-Based Photocatalytic Material

The phalsa fruit was purchased from a local market in Jamshoro, a city of Sindh, Pakistan. Then, the phalsa fruit was washed several times with the deionized water followed by the collection of juice using a juicer machine. The phalsa fruit is rich with the wide range of organic compounds including flavonols, dihydroflavonols, flavones, flavanols, anthocyanins, isoflavonoids, phenolic acids, flavanoes, and is most dominant inflavonols with 52.6% as shown by previous studies [88]. The pH of the fresh phalsa juice was observed at about 3.6 and we used 50 mL of phalsa juice placed in a 25 mL capacity beaker for the mild pyrolytic process at 220 °C for 2 h in the furnace with ramp rate of 12 °C min$^{-1}$. We obtained 3.5 g of carbon material, and it was dispersed in the deionized water followed by the sonication process for 30 min, and then filtered with ordinary filter paper. Later, the filtrate was treated with centrifugation at 4000 rpm for 25 min to remove the bigger carbon particles. At the end, we have successfully obtained 0.4 g of dark-reddish carbon particles. The preparation of carbon material from phalsa was pictorially presented in the Scheme 2 involving three stages: 1, mild pyrolytic process, 2, filtration, and 3, centrifugation. Synthesis of carbon material is shown in Scheme 2.

**Scheme 2.** Stepwise illustration of preparation process of carbon material from phalsa.

### 3.3. Physical Characterization of As-Prepared Carbon Material

We have characterized the morphology and crystal phase of the new carbon material prepared from phalsa fruit using scanning electron microcopy (SEM) (JSM-5910, JEOL Philips, Tokyo, Japan), and imaging was done at 20 kV and powder X-ray diffraction (PXRD) (Shimadzu-Model, Kyoto, Japan) using X-rays from the Cu anode with wavelength ($\alpha = 1.5406$ Å) operated 45 mA and 45 kV. The functional group study of the as-prepared carbon material was carried out with Fourier transform infrared spectroscopy (Tensor 27, Bruker Optics FT-IR, Karlsruhe, Germany). The optical studies were performed with the use of a UV-visible spectrophotometer (Lambda 365, PerkinElmer, Waltham, MA, USA) and absorbance was measured in the wavelength range of 200–800 nm.

### 3.4. The Photodegradation Application of Phalsa-Derived Photocatalytic Carbon Material

The methylene blue (MB) with an aqueous solution of $2.3 \times 10^{-5}$ M was prepared and various photocatalyst doses were used (5, 10 and 15 mg) and were placed separately in three beakers containing 50 mL of $2.3 \times 10^{-5}$ M MB. Then, carbon material was uniformly mixed with the dye solution using mechanical stirring for 30 min and stirring resulted in the formation of adsorption-desorption equilibrium in the dark on the surface of the carbon material. Afterwards, the photocatalyst, consisting of a dye solution, was placed in the locally made UV light box and the solutions were irradiated with UV light which enabled the activation of the newly-prepared carbon material. The UV light box contained six light emitting diodes (LEDs) of 365 nm wavelength and a power of 12 watt. The decrease in the absorbance was measured to ensure the oxidation of dye when UV light irradiated with the surface of carbon material, and then centrifuged, and 1 mL of each dye solution was kept in the Quartz cuvette cell of 1 cm and a change in the absorbance for several intervals of time were measured with a UV-visible spectrophotometer (Lambda 365, Perkin Elmer, Waltham, MA, USA). The absorbance of bare dye was recorded for the immediately prepared MB solution in deionized water. The observed absorbance at the maximum wavelength of 664 nm was connected to the typical molar absorptivity of MB. The photodegradation efficiency of newly-prepared carbon-based photocatalyst was estimated using following mathematical equation:

$$\text{Degradation Efficiency (\%)} = \left( \frac{A_o - A}{A_o} \right) \times 100 \tag{1}$$

Herein, $A_o$ represents absorbance of the initial dye concentration (mg/L), and A indicates the change in the concentration of dye (mg/L) after different intervals of time during the degradation process. The effect of pH on the degradation performance of the newly prepared carbon material was evaluated with a $2.3 \times 10^{-5}$ M dye solution and the pH of the solution was adjusted with the use of 0.2 M HCl and NaOH aqueous solutions. Furthermore, the scavenger studies were performed for an understanding of the nature of radicals involved in the degradation of MB under the illumination of UV light. For this purpose, we used different scavengers, including ascorbic acid ($C_6H_8O_6$), ethylenediamine tetracetate acid disodium (EDTA-Na2), and sodium monohydrate ($NaBH_4$) in the MB solution. The concentration of 10 mM for each scavenger was used and 60 µL was added to the MB solution using 15 mg of carbon material. The measurements of scavengers were performed with the irradiation of UV light. The role of active surface area and charge transfer was further studied by the cyclic voltammetry and electrochemical impedance spectroscopy (EIS). First, the electrochemical active surface area (ECSA) was measured through the non-Faradic region of CV curves at various scan rates in $2.3 \times 10^{-5}$ M. The electrochemical testing was performed with three electrode cell set ups, such as the reference electrode of silver-silver (Ag/AgCl) containing 3 M KCl solution, the platinum wire as counter electrode and the glassy carbon electrode (GCE) as working electrode. Prior to the electrochemical measurement, the GCE was polished with alumina paste and silicon paper and washed several times with the deionized water. Then, photocatalyst slurry of carbon material was made with the dispersion of 5 mg in 3 mL of deionized water and 100 µL of 5% Nafion solution. Later, 10 µL (0.2 mg) of photocatalyst ink was dropped on to GCE using the drop casting method. The modified electrode was dried at the room temperature. The surface area of GCE was about 3 mm. The EIS studies were done with a sweeping frequency of 100 kHz to 0.01 Hz and zero biasing potential and amplitude of 10 mV. The EIS was carried out in a $2.3 \times 10^{-5}$ M dye solution under the irradiation of UV light using an electrochemical workstation. The current density was calculated by the division of recorded current with the area of GCE. The EIS data was simulated with the well fitted equivalent circuit using Zview software for finding the charge transfer rate of carbon material during the degradation of MB. The size distribution of the carbon material was performed with the dispersed 3 mg of carbon material in 10 mL of the deionized water by employing Malvern Zetasizer Nano (ZS).

## 4. Conclusions

In summary, we have prepared carbon material from the phalsa fruit using a mild pyrolytic process under a limited air environment. The SEM, XRD, DLS, and UV-visible studies have shown the graphitic-like material with obvious carbon dot-like properties. The carbon material was potentially utilized against the degradation of MB in aqueous solution under the irradiation of UV light. Various photocatalytic parameters were studied, including the initial dye concentration, catalyst dose, the effect of pH of the dye solution, scavengers, reusability, and the charge transfer. It was observed that the degradation of MB was followed by the pseudo order kinetics and hydroxyl radicals were responsible for efficient degradation. Based on the obtained findings, it is clearly established that the proposed study for the production of photosensitive carbon material is considerable low-cost and provides access to high density surface sites, a minimum charge recombination rate, and, most importantly, can be scaled up for various applications such as bio-imaging, environmental applications, and energy conversion.

**Author Contributions:** Conceptualization, N.S.A.-K. and Z.H.I.; methodology, M.A.B.; validation, S.I.A.-S.; formal analysis, A.T.; investigation, G.M.A.-S.; resources, A.N. and Z.H.I.; data curation, Z.H.I.; writing—original draft, Z.H.I. All authors have read and agreed to the published version of the manuscript.

**Funding:** Not applicable.

**Data Availability Statement:** All research data is included in this article.

**Acknowledgments:** The authors express their gratitude to the support of Princess Nourah bint Abdulrahman University Researchers Supporting Project, number (PNURSP2023R58), Princess Nourah bint Abdulrahman University, Riyadh, Saudi Arabia. The authors extend their sincere appreciation to the Researchers Supporting Project, number (RSP2023R79), King Saud University, Riyadh, Saudi Arabia.

**Conflicts of Interest:** The authors declare no conflict of interest in the presented research work and agreed on the submission.

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
