# Peer review of "High Performance Carbon Material Prepared from Phalsa Using Mild Pyrolytic Process towards Photodegradation of Methylene Blue under the Irradiation of UV Light"

_catalysts, doi:10.3390/catal13020365_

Round 1
Reviewer 1 Report
Suggestion for Manuscript ID: catalysts-2216906
The present work, the authors used a mild pyrolytic process for the synthesis of luminescent carbon material from phalsa and utilized it for the photodegradation of MB in aqueous solution under the irradiation of UV light. propose the simple. This work proposed a low-cost and large-scale production of luminous carbon materials using phalsa. Over all, this manuscript was organized well and the research significance of this work was highlighted. However, there are still some questions that need to be appropriately revised before it can be published. My suggestions are as the follows:
1. Line 112-114, This expression should be modified to make it more reasonable, because the degradation efficiency is related to the reaction parameters.
2. Analytical methods, Chemicals and reagents must be provided in the paper.
3. Section 2.1, revised to “Preparation of luminescent carbon material” is better.
4. Section 2.1, the characterized for the morphology and crystal phase of new carbon material should be put into a single section: Instruments and analytical methods.
5. Line 165, the pH of solution was maintained with 0.2M HCl and NaOH aqueous solutions. Please revised to “the pH of solution was is adjusted with…” is better?
6. Line 338, 3.2.3, To keep the format consistent, Influence of pH of Mb dye should be revised to “Influence of pH on MB dye”.
7. If possible, the influence of inorganic ion (such as Cl-, HCO3-, PO43- et al.,) on MB removal should be discussed.
8. The references should be unified, such as Ref. [1]- [3], [6]-[10]. etc. without end pages.
Author Response
Comments and Suggestions for Authors
Suggestion for Manuscript ID: catalysts-2216906
The present work, the authors used a mild pyrolytic process for the synthesis of luminescent carbon material from phalsa and utilized it for the photodegradation of MB in aqueous solution under the irradiation of UV light. propose the simple. This work proposed a low-cost and large-scale production of luminous carbon materials using phalsa. Over all, this manuscript was organized well and the research significance of this work was highlighted. However, there are still some questions that need to be appropriately revised before it can be published. My suggestions are as the follows:
We are thank full to the reviewer for useful comments and suggestions in improving the quality of manuscript prior publication.
- Line 112-114, This expression should be modified to make it more reasonable, because the degradation efficiency is related to the reaction parameters.
Ans. Thank you for your comment and it has been acknowledged in the revised version of manuscript
- Analytical methods, Chemicals and reagents must be provided in the paper.
Ans. The analytical methods and chemical reagents have added in the revised version of manuscript
- Section 2.1, revised to “Preparation of luminescent carbon material” is better.
Ans. In the revised version section 2.1 has been revised.
- Section 2.1, the characterized for the morphology and crystal phase of new carbon material should be put into a single section: Instruments and analytical methods.
Ans. This has been done in the revised file
- Line 165, the pH of solution was maintained with 0.2M HCl and NaOH aqueous solutions. Please revised to “the pH of solution was is adjusted with…” is better?
Ans. It has been corrected in the revised manuscript
- Line 338, 3.2.3, To keep the format consistent, Influence of pH of Mb dye should be revised to “Influence of pH on MB dye”.
Ans. This has been modified in the revised manuscript
- If possible, the influence of inorganic ion (such as Cl-, HCO3-, PO43-et al.,) on MB removal should be discussed.
Ans. It has been previously shown that the inorganic anions like Cl-, HCO3-, PO43, , SO42-,etc have been accompanied by the textile industry waste water, hence they have significant effect on the retardation of MB dye oxidation. These kind of studies have to be considered during the oxidation of organic dyes and the role of such inorganic anions should be explored in order to realize the new generation of photocatalysts for the practical applications. This has been added in the revised manuscript.
- The references should be unified, such as Ref. [1]- [3], [6]-[10]. etc. without end pages.
Ans. The information about eh ending page, we could not find because at the each journal web page of such articles end page is not given, hence it is difficult to provide end page for each of them.
Reviewer 2 Report
The authors have synthesised a carbon-based catalyst and used it to degrade an aqueous solution of methylene blue.
Some major issues are as follows:
- In the abstract, the authors have mentioned carbon dots like properties of their materials, however, none of the carbon dots properties (size less than 10 nm, luminescence, ...) have been seen.
-Figure 7a and 7d -show the correlation coefficients
-Authors have talked about luminescence, however, the luminescent properties have not been studied. Also, these properties usually come from the recombination of the charge carriers which is not good for the photocatalytic application.
- Do the authors know about their carbon catalysts' porosity?
-What is the pHPZC of the catalysts, this can explain the degradation performances at different pH values.
-Page 4, line 162, please revise
-Concerning the EIS, it would be better to show them with and without light to shed light on the photocatalytic activity of the catalysts.
-Check table 2, in this work, it's not carbon dots. The SEM images as well as the DLS data do not show them.
-It would be better if the HPLC or TOC are carried out to have an idea about the intermediates generated or the mineralisation efficiency
Author Response
Comments and Suggestions for Authors
The authors have synthesised a carbon-based catalyst and used it to degrade an aqueous solution of methylene blue.
We are thank full to the reviewer for useful comments and suggestions in improving the quality of manuscript prior publication.
Some major issues are as follows:
- In the abstract, the authors have mentioned carbon dots like properties of their materials, however, none of the carbon dots properties (size less than 10 nm, luminescence, ...) have been seen.
Ans. This has been revised during revision
-Figure 7a and 7d -show the correlation coefficients
Ans. Both figures show the kinetics of degradation of MB under different condition. Figure 7a shows the kinetics study of MB degradation using different pH values of MB dye solutions. However, Figure 7d shows the kinetics of MB degradation under the influence of various scavengers. These corrections has been made in the revised manuscript
-Authors have talked about luminescence, however, the luminescent properties have not been studied. Also, these properties usually come from the recombination of the charge carriers which is not good for the photocatalytic application.
Ans. Yes, we did not study the luminescent properties of carbon material, however we have modify the luminescent word by using photocatalytic carbon material throughout the text of manuscript
- Do the authors know about their carbon catalysts' porosity?
Ans. We did not study the porosity of our carbon material.
-What is the pHPZC of the catalysts, this can explain the degradation performances at different pH values.
Ans. We did not calculate the pHPZC of the catalysts, however we have provided the shift in pH during the degradation of dye solution.
-Page 4, line 162, please revise
Ans., This has been revised
-Concerning the EIS, it would be better to show them with and without light to shed light on the photocatalytic activity of the catalysts.
Ans. Yes, the purpose, we studied the charge transfer through EIS was to see the kinetics of photocatalytic performance of carbon material under the illumination of UV light. At the present time, we face difficulty with experimental setup due to shift of laboratory, we have got issue instrument out of operation.
-Check table 2, in this work, it's not carbon dots. The SEM images as well as the DLS data do not show them.
Ans. This has been revised in the manuscript
-It would be better if the HPLC or TOC are carried out to have an idea about the intermediates generated or the mineralisation efficiency
Ans. The mineralization of MB into harmless products with different intermediate products have identified. However, in the present study, we aim to prepare the photocatalytic carbon material from green chemistry methodology, and we have evaluated the different parameters of MB dye degradation. Yes, we apologize for this additional information due to unviability of instrumentation in our lab.
Reviewer 3 Report
1. Some figures are with small and almost illegible legends on the graphics
2. “…stability and uniform dispersion in the polar solvents are the obvious features of biosynthesis.” This sentence could be updated the related work, such as Micropor. Mesopor. Mat, 341(2022) 112098 and Inorganics, 10(2022) 202. “The dye degradation in the presence of light has been governed and evaluated by many indicators for achieving high degradation efficiency and swift.” This could be added the current results, including CrystEngComm, 2022, 24, 6933–6943; Mater. Today. Commum., 2022, 31,103514 and ACS Appl. Mater. Interfaces., 2021, 13, 12463−12471.
3. Figure 8 is also with the caption very small and with low resolution. You must pass all graphics to Times as text and subtitles with readable and standard size on all.
4. Table 2 must be redone. Remove the features that are not used in scientific journals and improve its size. The header should be centered and not on multiple lines.
5. In the field of photocatalysis, the orbital energy of the semiconductor is very important, but in this paper, the author only calculate the Eg by the ultraviolet diffuse reflection. And it is recommended to supplement the orbital energy of the conduction band and valence band.
Author Response
Comments and Suggestions for Authors
We are thank full to the reviewer for useful comments and suggestions in improving the quality of manuscript prior publication.
- Some figures are with small and almost illegible legends on the graphics
Ans. The figures have been revised where necessary during the revision
- “…stability and uniform dispersion in the polar solvents are the obvious features of biosynthesis.” This sentence could be updated the related work, such as Micropor. Mesopor. Mat, 341(2022) 112098 and Inorganics, 10(2022) 202. “The dye degradation in the presence of light has been governed and evaluated by many indicators for achieving high degradation efficiency and swift.” This could be added the current results, including CrystEngComm, 2022, 24, 6933–6943; Mater. Today. Commum., 2022, 31,103514 and ACS Appl. Mater. Interfaces., 2021, 13, 12463−12471.
Ans. These corrections are made and the suggested citations are added.
- Figure 8 is also with the caption very small and with low resolution. You must pass all graphics to Times as text and subtitles with readable and standard size on all.
Ans. These corrections are made in the revised version of manuscript
- Table 2 must be redone. Remove the features that are not used in scientific journals and improve its size. The header should be centered and not on multiple lines.
Ans.table 2 is updated in the revised manuscript
- In the field of photocatalysis, the orbital energy of the semiconductor is very important, but in this paper, the author only calculate the Eg by the ultraviolet diffuse reflection. And it is recommended to supplement the orbital energy of the conduction band and valence band.
Ans. We understand the reviewer suggestions , however we don’t have expertise to calculate orbital energy of conduction and valence bands.
Round 2
Reviewer 2 Report
The revised manuscript can be accepted for publicatin
Reviewer 3 Report
published.